

# High-resolution terrestrial water storage dynamics in Central Asia: Evaluating hydrological forcing datasets for GRACE downscaling

Shuxian Liu[1], Timo Schaffhauser[2], Roland Pail[1]

[1]Chair of Astronomical and Physical Geodesy, Technical University of Munich, Munich, 80333, Germany
5    [2]Chair of Hydrology and River Basin Management, Technical University of Munich, Munich, 80333, Germany

*Correspondence to*: Shuxian Liu (shuxian.liu@tum.de)

**Abstract.** The Gravity Recovery and Climate Experiment (GRACE) and GRACE Follow-On (GRACE-FO) missions provide highly valuable large-scale observations of terrestrial water storage (TWS), but their coarse spatial (~200 km) and temporal (~monthly) resolutions limit their direct use in regional applications. In this study, we implement and refine a three-step downscaling framework to downscale GRACE-based TWS changes (TWSCs) to daily, 1 km resolution over the Naryn – Kara Darya basins and Fergana valley in Central Asia by integrating GRACE data with high-resolution hydrological forcing datasets, including precipitation, evapotranspiration, and runoff from Global Land Data Assimilation System (GLDAS), Famine Early Warning Systems Network Land Data Assimilation System Central Asia (FLDAS-CA), the land component of the Fifth Generation European ReAnalysis (ERA5-Land), and a mixed combination (Mix) comprising the Multi-Source Weighted-Ensemble Precipitation (MSWEP), the Global Land Evaporation Amsterdam Model (GLEAM), and the Global Flood Awareness System (GloFAS). Temporal downscaling corrects daily water-balance-derived storage changes using spline interpolation constrained by monthly GRACE observations. Spatial downscaling employs a Partial Least Squares regression to map coarse GRACE anomalies onto fine-scale predictors, and a post-bias correction ensures consistency with the original GRACE signal. Given the lack of in situ data in the study region, we implement three validation strategies: comparison with the ITSG-Grace2018 daily solution, an "upscaling-back" consistency test, and event-based analysis. The results show that all forcing scenarios capture the broad seasonal and interannual variability of GRACE, but their performance differs substantially. GLDAS retains grid-like artefacts, FLDAS-CA systematically underestimates long-term declines and seasonal amplitudes, and ERA5-Land introduces high-frequency noise in the daily TWSCs. In contrast, the Mix forcing achieves the best overall performance, yielding the highest correlation coefficients (up to 0.8) with ITSG-Grace2018, the most satisfactory Nash-Sutcliffe Efficiency (NSE) distribution (mean = 0.65) relative to GRACE signals in the upscaling-back test, and a realistic negative long-term trend (-5.7 mm yr$^{-1}$) compared with -8.3 mm yr$^{-1}$ from GRACE. The Mix-based downscaled product also captures short-term hydrological events, such as the significant January 2006 snow event, and human-induced impacts associated with potential return flows from surface water irrigation. These results highlight the importance of carefully selecting input hydrological datasets in downscaling applications. Additionally, the presented framework is computationally flexible and transferable, allowing specification of target resolutions and adaptation of input hydrological datasets accordingly for applications in other regions.



# 1 Introduction

Spatio-temporal analyses of terrestrial water storage (TWS) are fundamental to understanding how hydrological systems respond to climatic variability and human influences (Rodell et al., 2018; Tapley et al., 2019). TWS was adopted as a new Essential Climate Variable (ECV) by the Global Climate Observing System (GCOS) as part of the 2022 GCOS Implementation Plan. Over the past two decades, the Gravity Recovery and Climate Experiment (GRACE) and its successor, GRACE Follow-On (GRACE-FO), have provided observations of Earth's time-variable gravity field, offering invaluable insights into changes in TWS, which reflects the integrated signal in surface water, groundwater, soil moisture, snow, and glacier storages (Rodell and Famiglietti, 2001). These data have enabled global assessments of land water storage, ice mass balance, sea level change, and ocean bottom pressure, thereby improving our understanding of climate-driven Earth system variations (Tapley et al., 2019). However, GRACE's coarse spatial (~200 km) and temporal (~monthly) resolutions limit their direct application in regional hydrological studies.

To address this limitation, a range of downscaling approaches has been developed to enhance the spatial and temporal resolution of GRACE data. These methods are generally classified as dynamic, statistical, or machine learning-based. Dynamic downscaling employs physically based numerical models, e.g. regional climate or hydrological models, to generate high-resolution (HR) outputs while preserving underlying physical processes (Maraun et al., 2010). In contrast, statistical downscaling derives fine-scale estimates from coarse-resolution data by establishing relationships between large-scale predictors and local-scale predictands (Maraun et al., 2010). The latter is data-driven, computationally efficient, and thus widely used in GRACE applications. In recent years, machine learning techniques have been increasingly applied in GRACE downscaling to capture complex, non-linear relationships between large-scale gravity signals and high-resolution hydroclimatic variables.

Various studies have demonstrated the potential of such downscaling approaches. Vishwakarma et al. (2021) integrated GRACE observations with 0.5° × 0.5° water storage outputs from the Water – Global Assessment and Prognosis (WaterGAP) hydrological model (WGHM), combined with precipitation (P), evapotranspiration (ET), and runoff (R) data from multiple sources, to produce high-resolution TWS change (TWSC) fields. Yin et al. (2018) enhanced the spatial resolution of GRACE-derived groundwater storage anomalies in the North China Plain from 110 km to 2 km by incorporating ET data, thereby capturing sub-grid groundwater heterogeneity. Kalu et al. (2014) downscaled GRACE-derived TWSCs from 1.0° × 1.0° to 0.05° × 0.05° over a large hydrogeologic basin in northern Australia using P, ET, and R data from the Australian Water Outlook. Most recently, Pellet et al. (2024) developed a hybrid statistical dynamical approach to produce daily TWSC estimates at 1 km resolution by integrating GRACE data with auxiliary variables such as P, ET, and river network topography. Jyolsna et al. (2021) applied multi-linear regression (MLR) and random forest (RF) techniques to downscale GRACE-derived terrestrial water storage anomalies (TWSA) from 1° to 0.25°, using a range of land surface and hydroclimatic variables.

Despite these advances, limited attention has been paid to GRACE downscaling in high-mountain catchments of Central Asia. The southern Tien Shan, a critical water source for downstream regions, is characterised by runoff primarily derived from





snow and glacier melt (Pritchard, 2019; Immerzeel et al., 2020). Although hydrologically important, this area remains understudied in GRACE-based analyses. This region presents particular challenges due to its complex topography, strong seasonal variability (Barandun et al., 2021), scarcity of in-situ observations (Siegfried et al., 2012; Li et al., 2025), and strong upstream – downstream linkages (Pritchard, 2019). These factors make remote sensing data such as GRACE particularly valuable for hydrological assessment, yet also difficult to interpret at local scales without appropriate downscaling strategies.

Another crucial but often overlooked aspect concerns the choice of hydrological forcing datasets used for downscaling. Most existing studies rely on a fixed set of forcing variables, such as P, ET, or R, derived from reanalysis or land surface models (e.g. Arshad et al., 2022; Pellet et al., 2024; Kalu et al., 2024). However, the accuracy, resolution, and physical consistency of these datasets vary considerably, especially in topographically complex and data-scarce regions. As a result, different input hydrological data may introduce varying degrees of uncertainty into the downscaled TWS estimates, affecting their reliability

and hydrological interpretability. A systematic evaluation of how input hydrological forcing data selection influences downscaling performance is therefore essential but remains largely absent from current literature.

In this study, we downscale GRACE-based TWS changes to a target resolution of daily (temporally) and 1 km (spatially). This study aims to investigate the following research questions, on the example of a study region in Central Asia:

1.    How can GRACE-derived TWS changes be physically downscaled in space and time?

2.    To what extent and how does the choice of input hydrological forcing data affect the accuracy and robustness of downscaled GRACE products?

3.    How can the downscaled TWS products be validated in data-scarce regions?

4.    How can high-resolution TWS estimates improve our understanding of regional hydrological processes?

## 2 Study region

The study area is a transboundary catchment in Central Asia (Fig. 1), covering approximately 120,000 km². The dashed lines delineate three subregions, from east to west, corresponding to the Naryn Basin, Kara Darya Basin and the Fergana Valley, which straddle Kyrgyzstan and eastern Uzbekistan. The Naryn Basin, together with the Kara Darya, forms the Upper Syr Darya which is one of the two headwater catchments of the Syr Darya River in the Aral Sea Basin. The Naryn originates from the Big and Small Naryn rivers in the Tian Shan Mountains of Kyrgyzstan and flows westward though Fergana Valley into

Uzbekistan, where it joins the Kara Darya (Hagg et al., 2013). In this study, we refer to our transboundary catchment as the USD-FV. The basin is predominantly fed by glacio-nival rivers originating from high-altitude zones. Elevation ranges sharply from over 5,000 m in the glaciated peaks to around 200 m in the foothills. Precipitation, seasonal snow and glacier melt are the main sources of runoff, making this region crucial for sustaining downstream agriculture and water supply within the Syr Darya system (Zheng et al., 2019; Sadyrov et al., 2025).

This region exhibits pronounced climatic contrasts driven by its complex topography. The Upper Syr Darya Basin is characterized by a typical continental and semi-arid climate with hot summers and cold winters (Bocchiola et al., 2017). Mean



temperatures generally range between -18 °C and 17 °C (Zheng et al., 2019). In the Naryn Basin, the winter temperatures occasionally dropp below -50 °C and summer temperatures exceed 40 °C (Hill et al., 2017). Annual precipitation is relatively low, between 270 and 450 mm (Chen et al., 2022; Schaffhauser et al., 2023). In contrast, the Fergana Valley has a milder

continental climate with moderately cold winters and hot summers. Precipitation is particularly low, averaging only 100 – 200 mm yr$^{-1}$, whereas potential evapotranspiration can reach up to 1300 mm (Conrad et al., 2013). In addition, the valley is one of Central Asia's most important agricultural regions (Abdullaev et al., 2009).

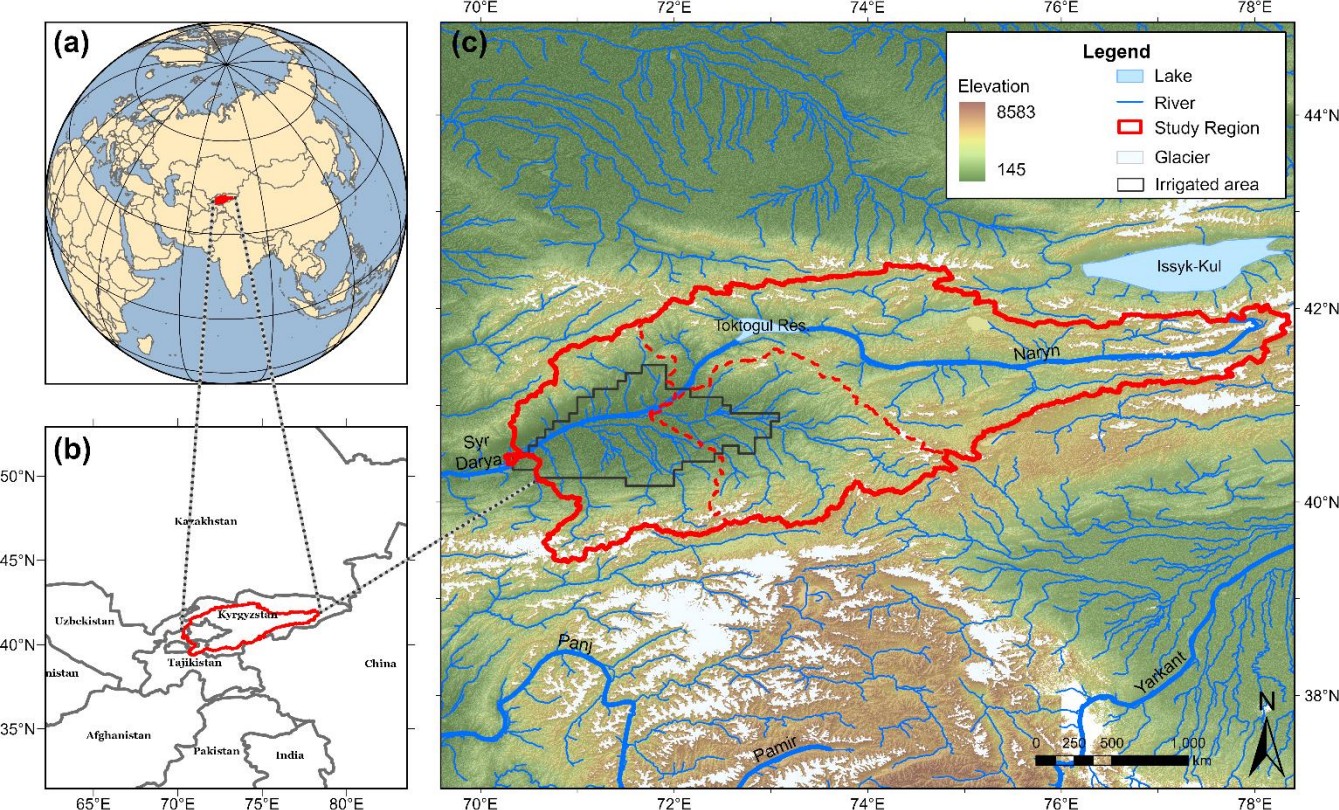

**Figure 1: Overview of the study region outlined in red, showing elevations, river networks, glaciers and lakes. The dashed lines**
**delineate three subregions from east to west: the Naryn Basin, the Kara Darya Basin, and the Fergana Valley. The black line delineates the irrigated area in the downstream region.**

## 3 Data

### 3.1 GRACE-based TWS

The GRACE and GRACE-FO missions provide monthly gravity field solutions disseminated by several processing centres at
different levels. Level-2 products are represented by spherical harmonic (SH) coefficients that require filtering and post-processing to obtain Level-3 gridded fields, typically provided at 1° × 1° resolution. An alternative is the mass concentration (mascon) approach, which directly estimates mass variations over predefined surface elements. Mascon solutions are produced



either directly from Level-1B data (e.g. NASA Goddard Space Flight Centre and Jet Propulsion Laboratory) or by post-processing Level-2 SH fields (e.g. Centre for Space Research).

We use the Jet Propulsion Laboratory (JPL) mascon monthly TWS product at 0.5° × 0.5° resolution for April 2002 – December 2023. Note that the JPL solution is originally estimated on ~3° equal-area blocks; the commonly used 0.5° grid is an interpolated version and does not add additional observational information (Watkins et al., 2015; Wiese et al., 2016). Within our study region, this corresponds to 74 0.5° grid cells, which are subsequently downscaled to the target resolution.

## 3.2 Hydrological datasets and other auxiliary information

Dynamical downscaling uses precipitation, evapotranspiration, and runoff at the target daily and 1 km × 1 km resolution. We evaluate four hydrological forcing sets, each comprising a combination of P, E, and R from the same model or multiple datasets, hereafter referred to as GLDAS, FLDAS-CA, ERA5-Land and Mix (Table 1). The following subsections summarise each dataset used in this study.

**Table 1: The four P – ET – R combinations used in this study.**

| Name | Precipitation | Evapotranspiration | Runoff | Temporal Resolution | Spatial Resolution |
|---|---|---|---|---|---|
| GLDAS | GLDAS | | | 3-hourly | 0.25° × 0.25° |
| FLDAS-CA | FLDAS-CA | | | Daily | 0.01° × 0.01° |
| ERA5-Land | ERA5-Land | | | Hourly | 0.1° × 0.1° |
| Mix | MSWEP | GLEAM | GloFAS | Daily | P/ET: 0.1° × 0.1°; R: 0.05° × 0.05° |

### 125 **3.2.1 GLDAS**

The Global Land Data Assimilation System (GLDAS) integrates satellite and ground-based observations with land surface modelling to provide globally consistent hydrological fields (Rodell et al., 2004b). We use P, ET, and R at 0.25° and 3-hourly resolution from the GLDAS Noah V2.1 model. Daily totals are obtained by summing 3-hourly values, then resampled to 1 km × 1 km using cubic interpolation (Arshad et al., 2022).

### 130 **3.2.2 FLDAS Central Asia**

The Famine Early Warning Systems Network Land Data Assimilation System (FLDAS) is a NASA-developed modelling framework designed for food and water security monitoring in data-scarce regions (McNally et al., 2022). FLDAS Central Asia model (FLDAS-CA) produces daily output including P, ET, and R at 0.01° spatial resolution over the Central Asia domain (30 to 100 °E, 21 to 56 °N).

### 135 **3.2.3 ERA5-Land**

The land component of the Fifth Generation European ReAnalysis (ERA5-Land; Muñoz-Sabater et al., 2021) is a global land-focused reanalysis dataset produced by the European Centre for Medium-Range Weather Forecasts (ECMWF). It provides

hourly land variables at 0.1°, forced consistently by ERA5. We derive daily P, ET, and R from the daily accumulated outputs (00:00 records) and resample to 1 km × 1 km.

### 3.2.4 MSWEP

The Multi-Source Weighted-Ensemble Precipitation (MSWEP) dataset combines gauge observations, satellite products, and reanalysis data to provide global precipitation from 1979 (Beck et al., 2019). We use daily data at 0.1°, interpolated to 1 km × 1 km.

### 3.2.5 GLEAM

The Global Land Evaporation Amsterdam Model (GLEAM) estimates terrestrial evapotranspiration and its components from remote sensing inputs combined with a simplified land surface model (Miralles et al., 2011). We use GLEAM V4 (Miralles et al., 2025) daily ET at 0.1°, resampled to 1 km × 1 km.

### 3.2.6 GloFAS

The Global Flood Awareness System (GloFAS), developed by the European Commission and ECMWF, is a global hydrological forecasting and monitoring system that couples meteorological forcing with a hydrological model to simulate river discharge and flood hazard (Alfieri et al., 2013). We use daily runoff fields at 0.05° from version 4.0 and resample them to 1 km × 1 km.

### 3.2.7 River network

The river network is derived from Hydrologic Derivative for Modelling and Analysis (HDMA) database developed by U.S. Geological Survey (Verdin, 2017), which provides digital elevation model (DEM)-based flow direction and flow accumulation at global scale. To match our target resolution, the flow direction grid is upscaled to 1 km.

### 3.2.8 Irrigation area

The Global Map of Irrigation Areas (GMIA; Siebert et al., 2013) provides the proportion of land equipped for irrigation around the year 2005 in percentage of the total area on a 5 minute resolution raster. This dataset is used to delineate the agricultural zone of the Fergana valley.

### 3.2.9 Overview of hydrological forcing datasets

Figure 2 provides an overview of the input hydrological variables used for GRACE TWS downscaling. From left to right, the columns show the average values of precipitation, evapotranspiration, and runoff for the period April 2002 – December 2023. The first three rows correspond to GLDAS, FLDAS-CA, and ERA5-Land, while the last row represents the Mix dataset





combining MSWEP, GLEAM, and GloFAS. All occurrences of "mm" in this manuscript refer to millimetres of water equivalent (mm w.e.).

Across all variables, ERA5-Land shows the highest magnitudes, while FLDAS-CA consistently yields the lowest. For precipitation, ERA5-Land exceeds 5 mm d$^{-1}$, whereas the other three datasets remain below 2 mm d$^{-1}$. Despite differences in magnitude, FLDAS-CA, ERA5-Land, and MSWEP identify similar wet hotspots, whereas GLDAS displays an unrealistic

spatial pattern. In particular, GLDAS indicates higher precipitation over the Fergana valley than in surrounding mountain regions, contradicting the well-established climatic knowledge (see Sect. 2), which shows the valley to be considerably drier. Among the four datasets, FLDAS-CA appears most realistic in both spatial distribution and magnitude. For evapotranspiration, ERA5-Land produces the largest values up to 2 mm d$^{-1}$, followed by GLDAS and GLEAM, while FLDAS-CA remains below 0.5 mm d$^{-1}$. The spatial pattern of ERA5-Land closely resembles GLEAM, whereas GLDAS and FLDAS-CA display elevated

ET over the valley region. GLDAS further exhibits grid-like artefacts akin to a random distribution. Notably, only FLDAS-CA shows an ET distribution that differs from its precipitation pattern, while in the other datasets, P and ET exhibit spatial consistency. In the GLEAM product, the narrow yellow band in the northeast denotes missing data over Song-Kul Lake. For runoff, ERA5-Land and GloFAS both reach values above 3 mm d$^{-1}$, whereas GLDAS and FLDAS-CA remain near 0.5 mm d$^{-1}$. The spatial distribution of runoff in ERA5-Land is broadly consistent with GloFAS, while GLDAS and FLDAS-CA lack

clear runoff hotspots.



**Figure 2: Spatial distribution of hydrological forcing variables used in this study, averaged over the study period. Columns represent (from left to right) precipitation, evapotranspiration, and runoff; rows correspond (from top to bottom) to the four forcing scenarios: GLDAS, FLDAS-CA, ERA5-Land, and Mix.**

# 4 Methods

## 4.1 Data pre-processing

GRACE products provide monthly terrestrial water storage anomalies relative to the long-term mean, defined as $TWSA = TWS - \overline{TWS}$. To analyse temporal variations in storage, we compute the time derivative of TWSA (dTWS/dt) in units of mm per month using a centred difference scheme applied to the JPL mascon data (Kalu et al., 2024):



$$\frac{dTWS}{dt}\Big|_{\text{month}} = \frac{dTWSA}{dt} \approx \frac{TWSA_{m_{i+1}} - TWSA_{m_{i-1}}}{2} = \frac{TWS_{m_{i+1}} - TWS_{m_{i-1}}}{2},$$ (1)

where $m_{i-1}, m_i, m_{i+1}$ denote three consecutive months. Forward and backward differences are used at the beginning and end of the time series, respectively.

Figure 3 outlines the workflow for GRACE data downscaling and validation of downscaled results. The process begins with data pre-processing, during which all input hydrological variables (P, ET, R) are resampled to 1 km × 1 km using cubic
interpolation. The flow direction dataset is upscaled to the same resolution based on the D8 algorithm, which assigns each grid cell a direction of deepest descent towards one of its eight neighbouring cells. Each direction is encoded by a unique power-of-two value: east (1), southeast (2), south (4), southwest (8), west (16), north (64), and northeast (128). The original 90 m resolution flow direction raster is aggregated to approximately 1 km using a block-wise mode approach. Firstly, an aggregation factor of 11 is defined since 1 km ≈ 90 m × 11. Next, the input raster is systematically partitioned into non-overlapping tiles of
11 × 11 pixels. At last, for each tile, the most frequent valid flow direction value is assigned to the corresponding upscaled pixel (NoData values are ignored). If all values are NoData, the output pixel is set to NoData. The affine transform is adjusted accordingly to yield a new pixel size of ~990 m.

Thereafter, a sparse connectivity matrix $Q \in \mathbb{R}^{n \times n}$ is constructed, where *n* denotes the number of 1 km × 1 km pixels within the study area. Each non-zero entry represents downstream connectivity:

$Q(p,q) = 1$      if flow from pixel *q* drains into pixel *p*. (2)

Daily changes in terrestrial water storage ($dS_t$, mm d$^{-1}$) are then derived from the water balance (WB) equation:

$$dS_t = \frac{dTWS}{dt}\Big|_{\text{day}} = P_t - ET_t - (I - Q) \cdot R_t$$ (3)

where $P_t, ET_t, R_t$ are precipitation, evapotranspiration, and runoff on day *t*, and *I* is the identity matrix.

Since daily fluxes are available, GRACE-based monthly TWS changes can be compared with WB-derived estimates.
Following Rodell et al. (2004) and Humphrey et al. (2023), monthly TWS changes based on WB-derived data are computed by Eq. (1) and

$$TWS_{m_i} = \frac{1}{d_e^{m_i} - d_s^{m_i} + 1} \sum_{d=d_s^{m_i}}^{d_e^{m_i}} \sum_{t=1}^{d} dS_t + TWS_{d=0}$$ (4)

where $d_s^{m_i}$ and $d_e^{m_i}$ denote the start and end days of month $m_i$.







**Figure 3: Workflow of GRACE data downscaling and evaluation of results.**

## 4.2 GRACE downscaling

### 4.2.1 Temporal downscaling

The downscaling of GRACE TWSA data is implemented through a three-step framework adapted from Pellet et al. (2024) and reorganized for clarity (Fig. 3). We provide a computation example in the Supplement. Step 1 involves temporal downscaling,

which aims to transfer the low-frequency GRACE signal onto high-frequency WB estimates and to produce a daily TWSC series on the GRACE-compatible grid for subsequent spatial downscaling. Starting from daily water storage changes ($dS$) at 1 km resolution derived from the WB equation, daily storage anomalies $S$ are reconstructed by cumulative summation. These $S$ values are then aggregated to the 0.5° GRACE grid by averaging across all 1 km pixels within each cell. The high-frequency series are segmented according to GRACE observation periods, and the mean value of each segment is computed for each grid

cell. Afterwards, the monthly differences between the JPL mascon TWSA and WB-based $S$ are interpolated in time using temporal splines to produce a daily correction term, which is added to the WB-based storage series to ensure temporal consistency with GRACE. At last, daily storage changes $dS$ at 0.5° are then obtained by numerical differentiation of the corrected daily storages. Note that these computations are carried out for each mascon pixel individually.

### 4.2.2 Spatial downscaling

Step 2 applies a Partial Least Squares (PLS) regression to relate GRACE-derived TWSA at coarse resolution to WB-based estimates at fine resolution. PLS is chosen for its ability to exploit spatial gradients and covariance structures across entire images. The regression establishes a statistical relationship between the temporally downscaled 0.5° TWSA and the 1 km WB-based predictors. The detailed steps can be found in Vishwakarma et al. (2021) and Pellet et al. (2024).

### 4.2.3 Post-bias correction

Step 3 ensures overall consistency between the downscaled high-resolution product and the original GRACE observations by correcting residual biases. The spatially downscaled daily $dS$ fields are first aggregated to the 0.5° grid, and the differences with the temporally downscaled $dS$ (Step 1) are quantified for each GRACE mascon and month. Following Pellet et al. (2024), the 1 km downscaled $dS$ is multiplied by land and river masks that are then aggregated separately to 0.5°. For each cell, the amplitudes of land- and river-based time series are used to determine their bias ratio. The resulting bias fields are spatially

interpolated to 1 km using a Radial Basis Function (RBF) approach and added to the downscaled product in Step 2. This yields the final high-resolution (1 km, daily) TWSC product that preserves GRACE's large-scale integrity while enhancing spatial detail for regional hydrological analysis.



### 4.3 Evaluation

Before downscaling, it is essential to assess the consistency between GRACE observations and WB-based ones. Figure 4
compares the basin-average monthly TWS changes from the JPL mascon product with water balance estimates derived from
four forcing scenarios (GLDAS, FLDAS-CA, ERA5-Land, and Mix) for 2002 – 2023. Panel (a) shows monthly TWSCs from
both sources, and panel (b) depicts their differences. All datasets reproduce the pronounced seasonal cycle of storage changes,
with positive values in spring and early summer and negative values in late summer to winter. However, the amplitude varies
substantially among datasets. GLDAS and Mix show the highest agreement with the JPL solution in both, phase and magnitude,
with correlation coefficients of 0.82 and 0.84 and root mean square errors (RMSEs) of 22.1 and 25.7 mm per month,
respectively. To place these error magnitudes in context, we also compute the mean total annual TWSC, defined as the mean
of the annual sums of absolute monthly TWSCs, which represents the typical annual amplitude of storage variation. The
resulting values are approximately 326 and 252 mm yr$^{-1}$ for GLDAS and Mix, respectively, compared to 317 mm yr$^{-1}$ for the
JPL mascon product. For ERA5-Land and FLDAS-CA, the corresponding values are 409 and 117 mm yr$^{-1}$. ERA5-Land
captures the seasonal dynamics but overestimates the peaks, resulting in a correlation of 0.75 and RMSE of 39.1 mm per
month, suggesting potential precipitation overestimation. FLDAS-CA performs weakest, with a dampened seasonal signal ($r$
= 0.48; RMSE = 29.4 mm per month) and systematic underestimation of seasonal extremes. Although GLDAS reproduces the
basin-mean TWS changes well, its unrealistic spatial distribution pattern particularly in precipitation (Fig. 2) warrants caution
in interpreting local results.



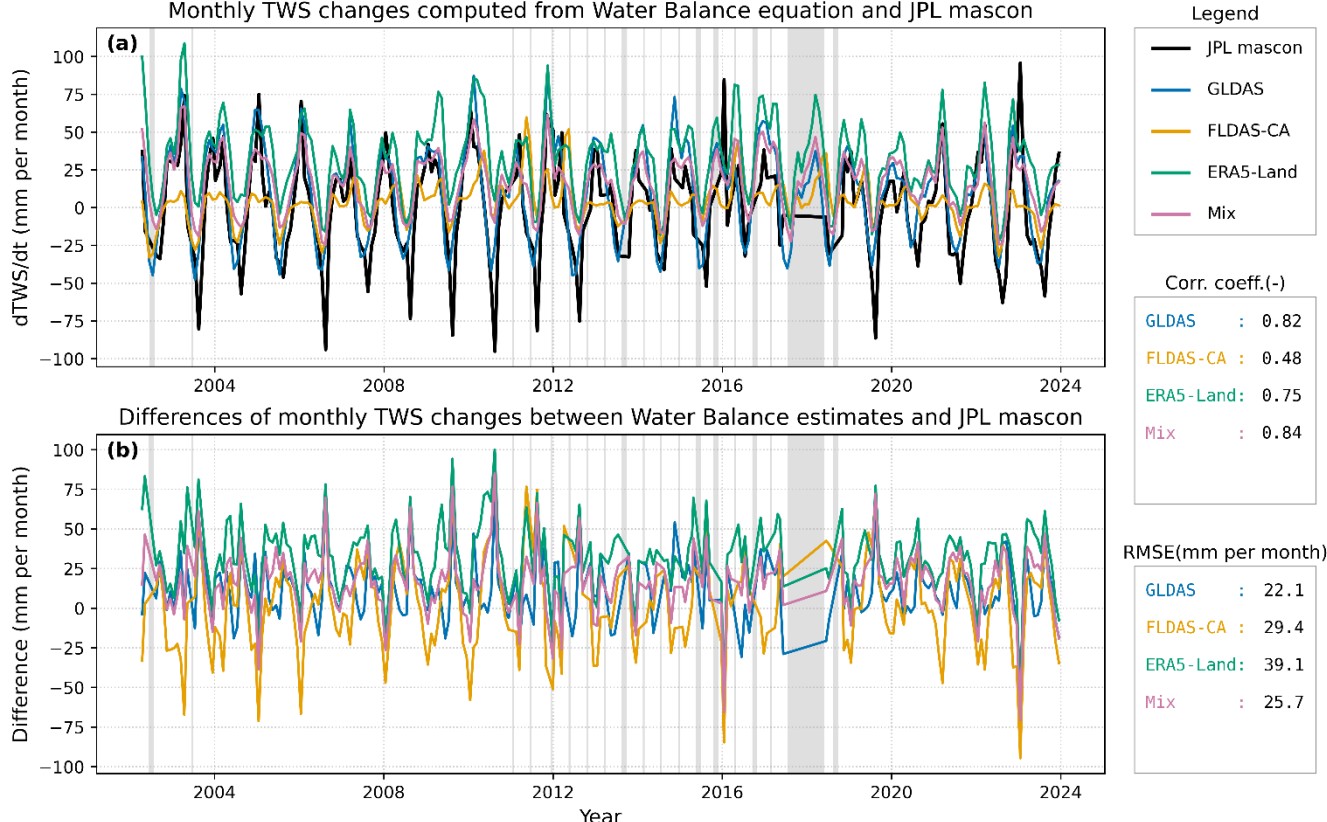

**Figure 4: Comparison of monthly TWSCs between April 2002 and December 2023 derived from four hydrological forcing scenarios (GLDAS, FLDAS-CA, ERA5-Land, and Mix) and the JPL mascon solution, averaged over the study region. Panel (a) shows monthly TWSCs computed from hydrological forcing datasets alongside the JPL mascon solution, while panel (b) presents their differences relative to JPL mascon. Grey-shaded areas indicate data gaps in the GRACE record. Correlation coefficients and root mean square errors with respect to the JPL mascon solution are shown on the right.**

Validation of GRACE downscaling results is often performed using independent in situ observations such as ground water level measurements. However, the lack of publicly available well data in this region prevents such comparison. Instead, three validation strategies are established as shown in the bottom right panel of Fig. 3. Firstly, the temporally downscaled products are compared with the ITSG-Grace2018 (hereafter ITSG2018) daily solution using the Pearson correlation coefficient and RMSE, which reveals systematic differences among the four forcing scenarios.

Second, an "upscaling-back" consistency test assesses whether the downscaled products preserve the large-scale variability of the original GRACE fields, thus ensuring mass conservation. Since GRACE measures time-variable gravity, consistency between the two datasets indicates that the downscaling process has not introduced spurious mass changes. Specifically, the downscaled 1 km daily TWSC grids are aggregated back to 0.5° resolution, after which monthly TWS changes are recalculated using Eqs. (1) and (4). The resulting "upscaled" TWSCs are then compared with the GRACE mascon-based results using the



Nash – Sutcliffe Efficiency (NSE) to quantify their agreement. As a widely used performance metric in hydrology, NSE measures how well the simulated values (here, the upscaled results) reproduce observed dynamics (GRACE mascon data), simultaneously considering both the mean and variance of the time series. An NSE value of 1 indicates perfect agreement, 0

signifies that the simulation performs no better than the long-term mean of the observations, and negative values imply poorer performance. Moreover, the long-term secular trends and seasonal amplitudes of TWS provide additional diagnostic validation. The linear trend and annual harmonic components are estimated by fitting the time series with offset, slope, and annual sine and cosine terms.

Third, event-based analyses can offer insight into the ability of the downscaled products to capture short-term hydrological

extremes in data-scarce regions. In addition, the downstream Fergana valley, which is extensively irrigated, warrants separate examination. Accordingly, this area is defined as the "highly irrigated area" (Fig. 1), while the remaining portion of the study region is referred to as the "non-irrigated area". The boundary between the two zones is delineated using the GMIA. It should be noted that the "non-irrigated area" does not imply a complete absence of irrigation but rather denotes where the proportion of irrigated lands is below 10%. Conversely, the "highly-irrigated area" refers to zones within the Fergana valley where more

than 70% of the land is under irrigation.

## 5 Results

### 5.1 Temporally downscaled TWSCs

In this section, the results of temporal downscaling (Sect. 4.2.1) are validated against the ITSG-Grace2018 daily solution. Figure 5 compares basin-averaged daily TWSCs (*dS*) derived from the four hydrological forcing scenarios (GLDAS, FLDAS-

CA, ERA5-Land, and Mix) with ITSG2018 for 2002 – 2023. Grey-shaded areas indicate periods when ITSG2018 data are unavailable (October – December 2016 and after mid-2017). Although the available ITSG2018 period is shorter than that of the input hydrological datasets, it still covers over 70% of the study period, providing a reliable basis for validating the temporally downscaled results. The ITSG2018 TWSCs mostly range between -5 and +5 mm d$^{-1}$. All downscaled datasets reproduce the seasonal variability observed in ITSG2018 but exhibit more pronounced seasonal fluctuations and differing

noise characteristics. GLDAS exhibits the highest variability, with frequent fluctuations exceeding 10 mm d$^{-1}$. While this indicates a high sensitivity to short-term forcing, it also suggests that GLDAS may introduce non-reasonable noise. In contrast, FLDAS-CA produces the lowest variability, yielding a smoother but reduced signal. Although this suppresses noise, it also appears to dampen genuine seasonal dynamics relative to ITSG2018. ERA5-Land lies between these two extremes, capturing seasonal cycles reasonably well but tending to produce stronger peaks than ITSG2018. The Mix scenario provides the most

balanced representation, closely matching ITSG2018 in both amplitude and variability while avoiding the excessive noise seen in GLDAS.

To quantitatively assess performance, the Pearson correlation coefficient and RMSE are calculated between the temporally downscaled TWSCs and ITSG2018 as a function of the averaging window (1 – 30 d) (Fig. 6a). For unsmoothed daily data,



RMSE values range from 2.5 to 3.2 mm d$^{-1}$ and correlations from 0.25 to 0.38, with Mix performing best and GLDAS worst.
Increasing the averaging window systematically reduces RMSE and improves correlations, indicating effective suppression of high-frequency noise and stronger consistency with ITSG2018 at coarser temporal scales. The most notable improvement occurs when applying a 5 d moving average, beyond which further smoothing yields marginal gains, implying that temporal downscaling introduces appreciable noise at the daily scale. Differences among the four scenarios diminish progressively with increased averaging. In the absence of smoothing, GLDAS and EAR5-Land yield substantially higher RMSEs compared with
the other two and also record the lowest correlation coefficients. By a 5 d window, ERA5-Land slightly surpasses FLDAS-CA in correlation. Ultimately, RMSE values stabilise near 0.75 mm d$^{-1}$ and correlations converge to approximately 0.77 across all scenarios when smoothing by a 30 d window. To further illustrate short-term dynamics, Fig. 6b presents a zoomed-in example for May 2010 – October 2011, showing basin-averaged daily TWSCs smoothed by a 10 d moving average. The time series of daily TWSCs from the four downscaling scenarios shows overall good agreement with the ITSG2018 solution. The
downscaled series align well with ITSG2018, reproducing major fluctuations and seasonal cycles, confirming that short-term smoothing effectively suppresses high-frequency noise. Nonetheless, certain negative peaks during spring 2011 observed in ITSG2018 are not fully captured, likely due to uncertainties in the input forcing data. Some discrepancies in amplitude remain, particularly, with FLDAS-CA and ERA5-Land occasionally exhibiting larger deviations.

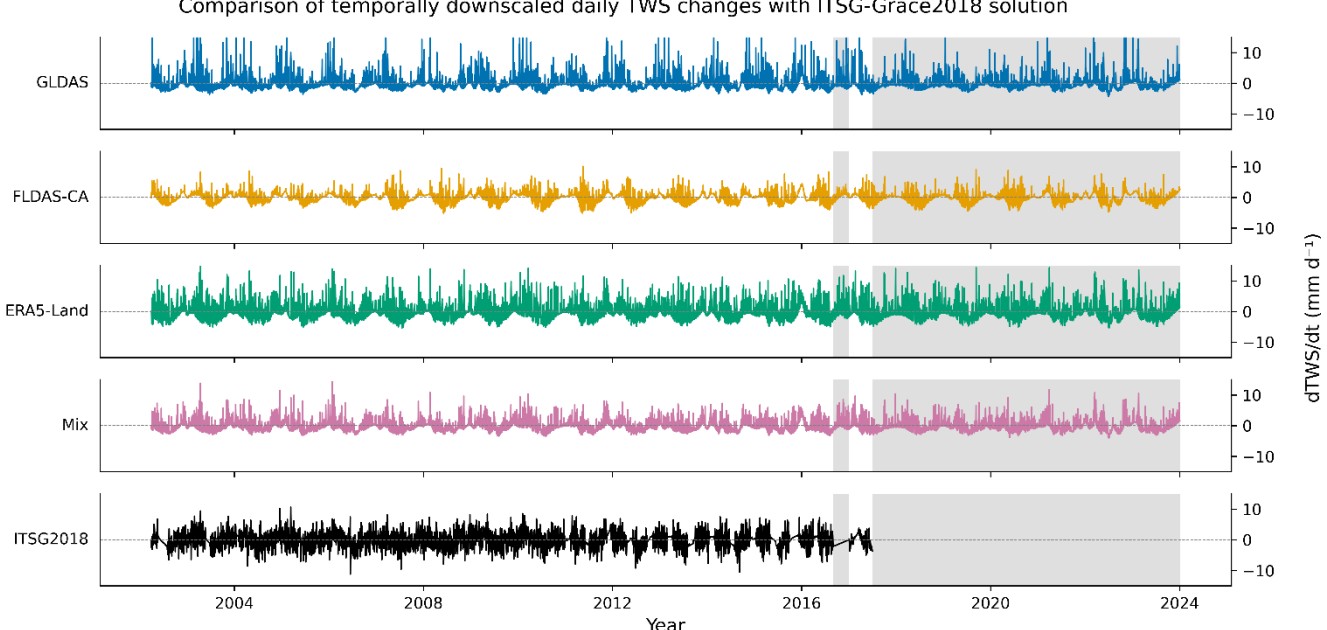

**Figure 5: Results of temporal downscaling (Step 1) compared with the daily ITSG-Grace2018 solution. Shown are basin-averaged daily TWSCs at 0.5° resolution, computed from the four hydrological forcing scenarios (GLDAS, FLDAS-CA, ERA5-Land, and Mix) and from ITSG-Grace2018. Grey-shaded areas indicate periods when ITSG-Grace2018 data are unavailable.**



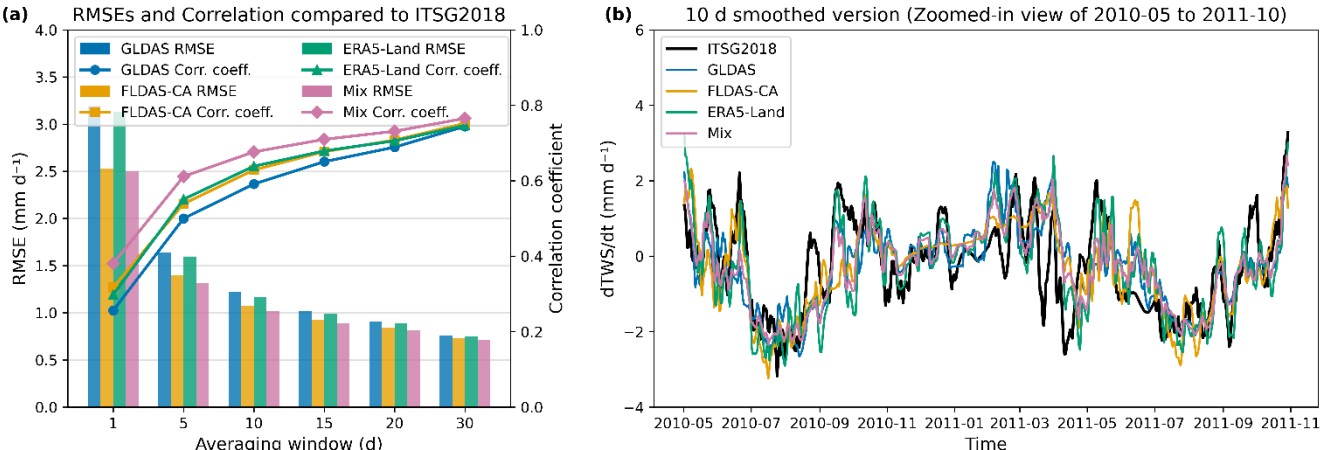

**Figure 6: (a) RMSEs (left axis) and correlation coefficients (right axis) between daily TWSCs derived from temporal downscaling and from the ITSG-Grace2018 solution. Both metrics are computed only for periods when ITSG-Grace2018 data are available. (b) Example time series of daily TWSCs between May 2010 and October 2011 from the four hydrological forcing scenarios and the ITSG-Grace2018 solution, smoothed using a 10 day moving window.**

## 5.2 Final downscaled results

High resolution (daily, 1 km) terrestrial water storage change (*dS*) products are obtained through applying PLS regression (Sect. 4.2.2) followed by post-bias correction (Sect. 4.2.3). Figure 7 illustrates, for 11 October 2023, the daily TWSC maps derived from WB-based, temporally downscaled, and final downscaled results across the four forcing scenarios (GLDAS, FLDAS-CA, ERA5-Land, and Mix). The WB-based *dS* maps (first column) exhibit fine-scale heterogeneity with patchy spatial patterns and marked differences among scenarios. ERA5-Land (g) and Mix (j) show more pronounced negative anomalies (dark red), suggesting stronger reductions in water storage. In contrast, GLDAS (a) and FLDAS-CA (d) display comparatively moderate changes closer to neutral (light red to light blue). Noisy local patches, particularly in the mountainous regions of ERA5-Land (g) and Mix (j), likely stem from uncertainties in runoff estimates under complex topography, where the insufficient representation of cryospheric processes often leads to imbalances in the water balance budget. The temporally downscaled dS maps (second column) appear spatially smoother, reflecting large-scale variations constrained by GRACE observations. Their magnitudes and spatial distribution differ from the WB-based fields because the temporal downscaling enforces GRACE consistency at coarse resolution. GLDAS and FLDAS-CA show relatively moderate daily changes, ERA5-Land exhibits a pervasive negative signal, and Mix provides an intermediate response. In the final downscaled results (third column), spatial details are restored, producing fine-scale patterns that merge large-scale trends with local variability. This demonstrates that the final downscaling effectively reconciles GRACE-derived temporal information with high-resolution hydrological features, though inter-dataset differences remain. Across all scenarios, a consistent progression is evident, from fine but unconstrained WB-based maps, to smoothed temporally downscaled fields, to spatially refined final products that reintroduce heterogeneity while retaining GRACE consistency. Notably, grid-aligned artefacts appear in GLDAS results,





manifesting as striping along model boundaries, reflecting the coarse native grid of GLDAS and potentially reducing fine-scale realism. ERA5-Land yields the strongest negative TWSC, whereas FLDAS-CA exhibits the weakest magnitudes, confirming that the final downscaled results remain highly sensitive to the choice of hydrological forcing dataset.

![Downscaled daily TWS changes on 11th Oct. 2023. Figure showing a 4x3 grid of spatial maps of daily TWSC (dS). Columns from left to right: WB-based dS, Temporally downscaled dS, Final downscaled dS. Rows from top to bottom: GLDAS (a,b,c), FLDAS-CA (d,e,f), ERA5-Land (g,h,i), Mix (j,k,l). Color scale dTWS/dt (mm d⁻¹) ranging from -2.0 to 2.0.]

**Figure 7: Spatial distribution of daily TWSC (*dS*) on 11 October 2023. Columns represent, from left to right, the WB-based estimates, temporally downscaled results, and finally downscaled results. Rows correspond, from top to bottom, to the four hydrological forcing scenarios: GLDAS, FLDAS-CA, ERA5-Land, and Mix.**






To evaluate the robustness of the final downscaled products, an "upscaling-back" consistency test (Sect. 4.3) is performed on
the 1 km daily TWSC estimates. Figure 8 uses October 2023 as an example to illustrate monthly TWS changes computed from
HR downscaled products in the four forcing scenarios and from the JPL mascon data. For each scenario, three panels are
shown: the TWSC grid derived from the HR downscaled product at 1 km (left), the upscaled field (middle), and the NSE map
between upscaled and mascon-based TWSCs over the entire study period (right). The JPL mascon field (panel a) reveals a
strong negative TWSC in the southern basin and moderate positive signals in the east. The HR downscaled products
successfully reproduce these broad spatial patterns while introducing fine-scale variability. However, positive anomalies occur
in the southern part of the FLDAS-CA result that are absent in the other products. The upscaled fields allow direct comparison
at the JPL mascon resolution. Across all scenarios, the main anomaly structures are generally consistent with JPL, though the
amplitudes vary depending on the forcing dataset. GLDAS and FLDAS-CA exhibit lower mass losses in the southern basin
compared with ERA5-Land and Mix, which both produce a stronger depletion of TWS. Nevertheless, all four scenarios
underestimate the magnitude of mass loss in the south relative to the JPL mascon product.

The NSE is classified into four categories to facilitate performance assessment, with values > 0.65 regarded as good, 0.50 –
0.65 as satisfactory, 0 – 0.50 as fair, and < 0 as poor (Moriasi et al., 2007). GLDAS performs well in most regions, with a
mean NSE of 0.68, but shows a localised area of low NSE in the central basin. Mix achieves predominantly high NSE values
(>0.65) across the study area, with a mean of 0.65 and no poorly performing zones, indicating strong agreement with JPL.
ERA5-Land and FLDAS-CA perform satisfactorily overall but exhibit several fair and poor regions, resulting in mean NSE
values of 0.55 and 0.50, respectively. It can be observed relatively low NSE values are concentrated in the central basin, which
is probably attributed to pronounced climatic and topographic contrasts between the surrounding high mountains and the low-
lying Fergana Valley. This transitional zone is characterised by steep elevation gradients, where temperature and precipitation
vary sharply over short spatial scales. Such heterogeneity increases the uncertainty of input datasets, particularly for
precipitation and evapotranspiration. Moreover, the coexistence of snow- and rain-dominated hydrological regimes enhances
the seasonal variability of storage changes, amplifying short-term mismatches between the downscaled and GRACE-based
TWSCs.







**Figure 8: Monthly TWSC for October 2023 derived from (a) the JPL mascon product and (b, e, h, k) the downscaled products obtained using different hydrological forcing datasets. Panels (c, f, i, l) show the upscaled TWSC fields aggregated from 1 km to 0.5° resolution based on the downscaled products. Panels (d, g, j, m) display the gridded NSE between the monthly TWSC time series derived from the upscaled results and the JPL mascon product.**

Figure 9 evaluates the consistency between the downscaled products and the JPL mascon data in terms of long-term TWS trends and annual amplitudes. Panel (a) presents the basin-averaged TWS time series from April 2002 to December 2023. All downscaled products reproduce the seasonal cycle and interannual variability observed in the JPL mascon solution. However, notable differences exist in long-term trends. The JPL mascon indicates a declining trend of -8.3 mm yr$^{-1}$, with pronounced losses after 2010. GLDAS and ERA5-Land closely follow this pattern, with trends of -6.1 and -6.0 mm yr$^{-1}$, respectively. The Mix product also captures a negative trend (-5.7 mm yr$^{-1}$), albeit weaker than JPL. In contrast, FLDAS-CA substantially underestimates the long-term decline, yielding only -4.1 mm yr$^{-1}$. The spatial patterns of TWS trends (panels b – f) further emphasise these differences. The JPL mascon trend map reveals strong negative trends concentrated in the southern basin. GLDAS, ERA5-Land and Mix broadly reproduce this spatial pattern with enhanced detail at 1 km resolution, whereas FLDAS-CA depicts a much weaker decline and fails to capture the pronounced negative signal in the south.

Panel (g) isolates the annual component of TWS, showing the amplitude and phase of the seasonal cycle. All downscaled products are consistent with JPL in phase, correctly reproducing the timing of recharge and depletion. Nonetheless, differences in amplitude exist. The JPL mascon yields a basin-averaged amplitude of 63.8 mm. GLDAS (56.7 mm) provides comparable though slightly weaker amplitudes, while ERA5-Land (49.2 mm) and Mix (47.3 mm) are moderate. FLDAS-CA again underestimates the signal considerably, with an amplitude of only 31.9 mm. The spatial distributions of amplitudes (panels h – l) confirm these results. JPL shows the strongest seasonal variation in the southern basin, which is captured with finer spatial detail by GLDAS, ERA5-Land and Mix, although the latter two tend to underestimate magnitude. FLDAS-CA produces the weakest amplitudes throughout the basin, suggesting a systematic dampening of seasonal variability. As noted previously, the noisy structures observed in the southern region are likely associated with glacierised areas in the high mountains, while the noise in the upper central basin may be linked to the Toktogul Reservoir, where complex and human-regulated water level changes dominate the local hydrological signal.





**Figure 9: (a)** Time series of TWS between April 2002 and December 2023 derived from the downscaled products in the four hydrological forcing scenarios (GLDAS, FLDAS-CA, ERA5-Land, and Mix) and from the JPL mascon solution. Dashed lines indicate the estimated long-term linear trends. Panels (b – f) show the spatial distribution of linear trends (mm yr⁻¹) estimated from the JPL mascon product and the four downscaled datasets. The numbers in the lower right corners denote the basin-averaged secular trends corresponding to panel (a). Panel (g) presents the annual components of the TWS time series from the same datasets. Panels (h – l) display the gridded amplitudes (mm) estimated from the JPL mascon product and the downscaled results.

## 5.3 Analysis in the irrigated area

Based on the analyses presented above, the Mix downscaled product, based on the combination of MSWEP, GLEAM, and GloFAS), demonstrates the best overall performance in capturing temporal variability and maintaining consistency with the large-scale GRACE signals. Therefore, the Mix product is selected for subsequent sub-basin spatio-temporal and event-based analyses.

The downstream basin is mainly occupied by irrigation identified in Fig. 10a (same as Fig. 1). Figure 10 illustrates the differences in TWS dynamics between the highly irrigated and non-irrigated regions. Panel (b) compares the basin-averaged TWS time series from 2002 to 2023. Both regions exhibit a pronounced seasonal cycle and an overall declining long-term
425 trend, but the irrigated area shows a systematically weaker TWS reduction than the non-irrigated one. Panels (c) and (d) show daily TWS changes from the Mix downscaled product alongside daily precipitation in irrigated and non-irrigated areas, respectively. In both regions, the correlation coefficients between P and TWSC are approximately 0.9, confirming the dominant role of P in driving short-term water storage variations. On average, precipitation in the highly irrigated area is roughly half that in the non-irrigated region, while evapotranspiration magnitudes are comparable between the two. Nevertheless, clear
430 differences emerge in the long-term TWS behaviour. In the non-irrigated area (Fig. 10d), negative TWSC values are more frequent and persist longer, particularly during summer and early autumn. In contrast, in the highly-irrigated area (Fig. 10c), the seasonal decline is less severe, suggesting a potential contribution from surface irrigation return flow that partially offsets storage losses.







**Figure 10: (a) The location of highly irrigated areas within the study region. (b) Time series of TWS averaged over highly irrigated and non-irrigated areas between April 2002 and December 2023. (c, d) Daily TWSCs (left axis) from the Mix downscaled product and precipitation (right axis) in highly irrigated and non-irrigated areas, respectively.**

There is a distinct positive peak observed in both regions at the beginning of 2006, as shown in Fig. 10c and Fig. 10d, corresponding to the day of maximum TWS increase and precipitation. Figure 11 illustrates the spatial distribution of daily TWS changes and precipitation from 25 and 30 January 2006. According to ReliefWeb (2006), a strong snowfall event occurred between 26 and 28 January 2006, which is clearly reflected in both the precipitation fields (Part II, panels h – j) and the downscaled daily TWS changes from the Mix product (Part I, panels b – d). On 26 January, the onset of snowfall produced moderate precipitation across the basin, accompanied by positive TWSC values of $5 - 10$ mm d$^{-1}$ in the central and southern regions. The event peaked on 27 January, when heavy snowfall exceeded 25 mm d$^{-1}$ over the southern basin (panel i), and the TWSC response surpassed 20 mm d$^{-1}$ (panel c), indicating rapid accumulation of water mass within the snowpack. Residual positive anomalies persisted on 28 January (panels d and j), reflecting continued storage of snow water equivalent even after the main precipitation had subsided. By $29 - 30$ January, both precipitation and storage changes returned to near-background levels.

Meteorological records from ground stations (Menne et al., 2012) corroborate this event (Fig. 12), which shows daily precipitation at the FERGANA (40.4 °N, 71.8 °E) and ANDIZAN (40.8 °N, 72.3 °E) stations between January and February 2006. Both stations record peaks exceeding 20 mm on 27 January 2006, consistent with our analysis from the downscaled results. This case study highlights the ability of the downscaled GRACE product to capture short-term hydrological responses to extreme events. The strong spatial and temporal correspondence between the recorded snowstorm and the positive TWS changes demonstrates that the Mix downscaled product effectively detects daily water mass variations associated with snow accumulation. In contrast, GLDAS fails to reproduce the event, while FLDAS-CA captures a TWSC peak on 27 January consistent with the Mix product (Fig. 11), but with a much lower magnitude. ERA5-Land, by contrast, shows a one-day delay, with the TWSC peak occurring on 28 January and being spatially displaced (see Supplement). Across all products, precipitation signals are mirrored by TWSCs, confirming that short-term TWS variability is primarily driven by precipitation, and reinforcing the central role of hydrological forcing data quality in determining downscaling performance.







Figure 11: Daily TWS changes (Part I) from the Mix downscaled product and corresponding precipitation patterns (Part II) for the period 25 – 30 January 2006.







**Figure 12: Precipitation records from the meteorological stations between 1 January and 25 February 2006. The inset in panel (b)**
**indicates their locations within the study region.**


## 6 Discussions

This study demonstrates both the potential and limitations of integrating GRACE observations with high-resolution hydrological forcing datasets to produce daily and kilometre-scale estimates of terrestrial water storage changes. A key factor influencing the final downscaled product is the choice and quality of input data, including both the GRACE product and the

input hydrological forcing data. To minimise uncertainties related to post-processing of GRACE data, this study employs the JPL mascon solution as input rather than GRACE Level-2 SH coefficients. The downscaling framework successfully preserves the coarse-scale integrity of GRACE while enhancing spatial and temporal detail. However, several sources of uncertainty remain in this process. Firstly, the use of high-resolution forcing data may introduce both meaningful daily variability and undesired high-frequency noise. Secondly, the relationship between coarse and fine-scale variables may also be partially

captured by the predictors used in PLS regression. Additional uncertainty arises from imperfect water balance closure, as



inconsistencies among hydrological datasets, anthropogenic influences, and complex surface – subsurface interactions can cause the simplified water balance equation to deviate from actual storage dynamics. For instance, in the study region, irrigation relies primarily on surface water from precipitation, snow and glacier melt rather than groundwater extraction, which affects local recharge processes. Despite these potential inconsistencies, their fine-scale spatial information from hydrological forcing

datasets remains valuable for incorporating high-frequency variability and spatial distribution on smaller scales in the GRACE data.

Previous studies have explored various strategies for GRACE downscaling, spanning physically based dynamic models to data-driven statistical and machine learning approaches. Most existing work focuses on single spatial downscaling schemes (e.g. Kalu et al., 2014; Yin et al., 2018), whereas only a few studies have attempted to combine dynamic and statistical methods

within a unified framework to exploit their complementary strengths and limitations. The framework adopted in this study follows a hybrid statistical – physical concept similar to Pellet et al. (2024), but we reorganise the process into a clearer, stepwise workflow and provide explicit computational guidance (see Supplement). In contrast to most previous studies (e.g. Arshad et al., 2022; Pellet et al., 2024), which generally applied a single forcing dataset (typically GLDAS or ERA5) for downscaling, our analysis systematically evaluates the influence of multiple hydrological forcing datasets. This inter-

comparison reveals the sensitivity of downscaled results to input data quality and the representation of surface fluxes. Beyond magnitude differences, our results demonstrate that inconsistencies in forcing data can alter the spatial coherence, temporal smoothness, and overall stability of the derived water storage changes. Although studied for the Naryn – Kara Darya basins and Fergana valley, the framework is transferable to other data-scarce and topographically complex regions. It allows substitution of different forcing datasets or regional hydrological models, provided that daily flux estimates and flow direction

information are available.

Validation of downscaled GRACE products remains a major challenge, particularly in data-scarce regions such as Central Asia. Some studies have incorporated ground-based observations, such as well water levels or river discharge, to evaluate local performance (Li et al., 2019; Arshad et al., 2022). However, these quantities are often obtained through signal separation of downscaled TWS, which introduces additional uncertainties, as the non-groundwater components used for isolation are

typically derived from model simulations with their own biases. In our study region, where reliable in situ observations are unavailable, the adopted validation framework provides a balanced and pragmatic assessment of the downscaled results in terms of temporal, spatial, and process-based consistency. This strategy offers a feasible way for evaluating GRACE downscaling performance in regions where direct ground validation cannot be achieved.

In addition, the rapid progress in GRACE downscaling research offers great potential for strengthening the link between

experimental developments and practical hydrological applications. Much of the recent work, driven by the GRACE and remote sensing communities, has provided valuable advances in algorithmic refinement and methodological intercomparison. Building on these achievements, future efforts could further enhance the integration of downscaled GRACE products into operational hydrological models, drought monitoring systems, and water management frameworks.



**7 Conclusions**

This study applies a three-step framework to downscale GRACE and GRACE-FO terrestrial water storage data to daily, 1 km resolution across the Upper Syr Darya – Fergana basin in Central Asia. The method integrates GRACE observations with high-resolution hydrological forcing datasets of precipitation, evapotranspiration, and runoff from four scenarios: GLDAS, FLDAS-CA, EAR5-Land, and a mixed combination (MSWEP + GLEAM + GloFAS), through temporal spline correction, spatial regression, and post-bias adjustment. The downscaling framework effectively preserves the coarse-scale integrity of GRACE

while reconstructing physically consistent, high-resolution fields that reveal sub-mascon hydrological variability.

The hydrological forcing information exhibits distinct behaviours in P, ET, and R, underscoring the importance of dataset selection. Among them, ERA5-Land consistently produces the highest magnitudes (mean P, ET, and R up to 5, 2, and 3 mm $d^{-1}$, respectively), especially tending to overestimate P, while FLDAS-CA provides the lowest values (< 2, 0.5, and 0.5 mm $d^{-1}$). In terms of spatial patterns, GLDAS displays unrealistic patterns in P and ET, whereas FLDAS-CA tends to dampen spatial

variability, particularly for ET and R. These inconsistencies directly affect the realism of water balance closure and the resulting downscaled outcomes. The water balance-based TWSCs using P, ET, and R reproduce the seasonal and interannual variability of GRACE, with correlation coefficients of 0.82 and 0.84 and RMSEs of 22.1 and 25.7 mm per month for GLDAS and Mix, respectively. FLDAS-CA and ERA5-Land yield lower correlations (0.48 and 0.75) and higher RMSEs (29.4 and 39.1 mm per month), which demonstrates the strong dependence of downscaling accuracy on the choice of hydrological forcing.

Given the scarcity of ground observations in the study region, three validation approaches are implemented: (i) comparison with the ITSG-Grace2018 daily solution, (ii) an upscaling-back consistency test, and (iii) event-based evaluation. The four forcing scenarios show clear performance differences. In the comparison with ITSG2018, GLDAS exhibits the weakest agreement with $r = 0.25$ and RMSE = 3.2 mm $d^{-1}$, while Mix achieves the best performance with $r = 0.38$ and RMSE = 2.5 mm $d^{-1}$. FLDAS and ERA5-Land performs comparably with the correlation coefficient of approximately 0.3, though ERA5-

Land yields higher RMSE (3.2 mm $d^{-1}$) with respect to FLDAS-CA (2.5 mm $d^{-1}$). Increasing the smoothing window improves all scenarios, with RMSE stabilising near 0.75 mm $d^{-1}$ and correlations converging to approximately 0.77 across all scenarios at a 30 d window, indicating that temporal downscaling introduces appreciable high-frequency noise. In the "upscaling-back" test, GLDAS and Mix perform well with mean NSE values of 0.68 and 0.65, followed by ERA5-Land (0.55) and FLDAS-CA (0.5). Mix is the only one without "poor" cells in the gridded NSE map, while FLDAS-CA produces the smallest number of

"good" grid cells. In terms of the long-term TWS trend, the estimates derived from the downscaled products reveal consistent basin-wide depletion with -6.1, -4,1, -6.0, and -5.7 mm $yr^{-1}$ for GLDAS, FLDAS-CA, ERA5-Land, and Mix, compared with -8.3 mm $yr^{-1}$ from GRACE observations. Across all datasets, differences in long-term trend and seasonal amplitude reach approximately 20%, underlining the importance of carefully selecting or combining input hydrological forcing datasets for GRACE downscaling, particularly in topographically complex and data-scarce regions. Among our tested data, FLDAS-CA

reproduces the temporal variability reasonably well but underestimates long-term changes with a dampened amplitude of nearly 50%. Both GLDAS and ERA5-Land introduce much high frequency noise. However, when aggregated to the basin





scale, GLDAS performs well, despite its unrealistic grid-like spatial artefacts. ERA5-Land tends to exaggerate seasonal peaks and introduce excessive short-term fluctuations. The Mix dataset achieves the best balance, with highest correlation and lowest RMSE relative to ITSG2018, no "poor" cells in the "upscaling-back" test, and a realistic representation of the negative long-term trend.

The downscaled products capture both long-term and short-term hydrological processes. In the Mix downscaled result, seasonal cycles are well represented, and the framework successfully reproduces short-term hydrological events such as the January 2006 snowstorm, where strong precipitation and snowfall were reflected as rapid positive TWSC anomalies. Furthermore, human-induced influences are also evident when comparing highly irrigated and non-irrigated areas. Although quantification and validation require further investigation, the observed differences in TWS dynamics suggest that irrigation mitigates the magnitude of seasonal storage decline, likely due to surface water return flows.

The results of this study demonstrate the potential of GRACE downscaling for high-resolution water storage monitoring in data-scarce and topographically complex regions. In the future, several aspects warrant further improvement. First, the accuracy of the downscaled products remains constrained by the quality and physical consistency of the forcing datasets. Ensemble-based or data-assimilation approaches that integrate multiple P, ET, and R products could mitigate input-related uncertainties. Second, explicit representation of snow and glacier processes is required to better capture cryospheric contributions, which are critical in high-mountain Central Asia. Third, the next-generation gravity mission, the Mass Change and Geophysics International Constellation (MAGIC) will provide higher-resolution gravity field data, which offers new opportunities to improve the downscaling accuracy.

The framework presented in this study can be easily generalized to other regions, and the computation example provided in the Supplement offers region-independent computational procedures. It is computationally flexible and transferable. Users can specify their own spatial or temporal target resolutions (e.g. a target spatial resolution as 0.25°), and the input hydrological forcing datasets can be replaced with variables from other sources, provided that their resolution matches the target resolution. On the other hand, users may also choose to perform only temporal or spatial downscaling, depending the research objectives. In such cases, the input hydrological forcing data should be adjusted accordingly. For instance, when applying temporal downscaling alone, the hydrological variables must have the same temporal resolution as the target temporal resolution, while their spatial resolution need not match that of the GRACE data. Conversely, when applying spatial downscaling only, the hydrological variables should be consistent with the target spatial resolution but retain the same monthly temporal resolution as the GRACE data.

**Data availability**

The JPL GRACE mascon product (Wiese et al., 2023) used in this study is available at https://doi.org/10.5067/TEMSC-3JC634. The GLDAS Noah V2.1 data (Beaudoing and Rodell, 2020) is available at https://doi.org/10.5067/E7TYRXPJKWOQ. The FLDAS-Central Asia product (Slinski and Sarmiento, 2023) is available at



https://doi.org/10.5067/C4IOYF41EEZB. The ERA5-Land hourly data (Muñoz-Sabater, 2019) are available at https://cds.climate.copernicus.eu/datasets/reanalysis-era5-land?tab=overview. The MSWEP product (Beck et al., 2019) is available at https://www.gloh2o.org/mswep/. The GLEAM4 datasets (Miralles et al., 2025) are available at https://www.gleam.eu/. The runoff data by GloFAS (Grimaldi et al., 2022) is available at https://doi.org/10.24381/cds.ff1aef77. The river flow direction data (Verdin, 2017) is available at https://doi.org/10.3133/ds1053. The irrigation maps of GMIA (Siebert et al., 2013) are available at

https://www.fao.org/aquastat/en/geospatial-information/global-maps-irrigated-areas/latest-version/. The daily precipitation data (Menne et al., 2012) for the FERGANA and ANDIZAN stations are obtained from https://www.ncei.noaa.gov/products/land-based-station/global-historical-climatology-network-daily. The GRACE downscaled TWSC estimates in this study (Liu et al., 2025) are available at https://doi.org/10.5281/zenodo.17466845.

**Author contribution**

SL and TS designed the experiment; SL performed the computations, analyses, and visualization; TS and RP contributed to conceptualization and data analysis; SL wrote the manuscript draft; TS and RP reviewed and edited the manuscript. RP supervised the study.

**Competing interests**

The authors declare that they have no conflict of interest.

**Acknowledgements**

This study was funded by the TUM Innovation Network "Twin Earth Methodologies for Biodiversity, Natural Hazards, and Urbanization (Earthcare)", Technical University of Munich.

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
