# Peer review of "High-resolution terrestrial water storage dynamics in Central Asia: Evaluating hydrological forcing datasets for GRACE downscaling"

_EGUsphere, 2025_

## Referee Comment (RC1)

**General Comments**

This study focuses on the downscaling of terrestrial water storage change (TWSC) derived from the Gravity Recovery and Climate Experiment (GRACE) and GRACE Follow-On (GRACE-FO) missions to a daily temporal resolution and 1 km spatial resolution over the Naryn–Kara Darya basins and the Fergana Valley in Central Asia. The authors implement and refine a three-step downscaling framework that integrates GRACE observations with high-resolution hydrological forcing datasets of different natures (reanalysis- or observation-based), namely precipitation (P), evapotranspiration (E), and runoff (R). The objectives of the study are threefold: (i) to evaluate the sensitivity of the downscaling accuracy to different hydrological forcing datasets, (ii) to develop validation strategies in data-scarce regions, and (iii) to demonstrate the added value of high-resolution TWSC products for improving hydrological process understanding.

The manuscript addresses relevant scientific questions within the scope of *Hydrology and Earth System Sciences* (HESS) and evaluates recent methodological advances in GRACE downscaling. The paper present important bottleneck: the use of input in the physical downscaling while addressing indirect validation. The methods and assumptions are valid and overall presentation is well structured and clear. While the study shows potential, major revisions are required to meet the standards of the journal. In particular, the manuscript suffers from issues related to notation consistency, dataset description, methodological clarity, and the depth of the evaluation. The comments below are intended to help the authors improve the manuscript accordingly.

**Major Comments**

**1. Notation and Conceptual Clarity**

- There are several instances of ambiguity in the notation used throughout the manuscript, particularly regarding: Total Water Storage (TWS), Total Water Storage Anomalies (TWSA), Total Water Storage Change (TWSC), and alternative notations such as *S*, *dS*, *dTWS/dt*, "TWS change" etc. This lack of consistency makes it difficult to follow the manuscript. I recommend adopting a single, consistent notation throughout the manuscript, including the main text, equations (Eqs. 1–4), and figure labels. Using either: TWSA / TWSC, or S / dS. A careful revision of the entire manuscript is required to resolve these ambiguities.

**2. Dataset Description**

- The ITSG-2018 dataset used for evaluation is not clearly introduced and should be described in Section 3.1, alongside the GRACE Mascon dataset. Furthermore, ITSG-2018 is available for periods beyond 2017 and can be downloaded from:
https://ftp.tugraz.at/outgoing/ITSG/GRACE/ITSG-Grace_operational/daily_kalman/netcdf/

- The hydrological datasets (Section 3.2) are insufficiently described. In particular, the nature of each dataset (observed, reanalysis-based, or model-derived) should be clearly stated, especially for precipitation and evapotranspiration. For evapotranspiration, the underlying model(s) should be specified. This information is crucial when discussing the limitations and uncertainties associated

with different forcing combinations and directly relates to the core objectives of the study. Also it is not clear how GLOFAS runoff differ from ERA5 runoff.

-The authors should better emphasize that the MIX combination relies primarily on satellite observations, whereas the other combinations do not, which could explain why the downscaling performs better in this case. Indeed, MSWEP aggregates multiple datasets, several of which are based on Earth observations; GLEAM is derived from satellite and reanalysis data.

3. Methodological Issues

- In line 205, it is unclear whether the connectivity matrix Q accounts only for direct upstream ("parent") pixels or for all upstream contributors (i.e., the entire upstream catchment). This distinction is critical, as it determines whether runoff exchanges are evaluated locally or integrated over the full drainage area. In addition, the methodology implicitly treats runoff as a proxy for discharge. However, water mass conservation in hydrology is governed by discharge, not runoff. Considering daily runoff exchanges between adjacent pixels implies assumptions about water travel time (e.g., 1 km per day), whereas aggregating runoff over the full upstream area implicitly treats runoff as a net flux at the pixel scale. These assumptions should be explicitly stated and discussed in the manuscript.

- In the temporal downscaling step (Section 4.2.1), it is unclear how months with missing GRACE TWSC data prior to 2017 are handled. Figure 5 does not indicate these gaps, although they exist. Moreover, the time series suggests that linear interpolation at the monthly scale may have been applied prior to temporal downscaling (e.g., in 2017). I recommend masking all months with missing GRACE data rather than interpolating them, and clearly documenting this choice in both the methodology and the figures.

- The description of the spatial downscaling procedure (Section 4.2.2) should be expanded as the manuscript should provide a self-contained explanation of the main principles and assumptions underlying the spatial disaggregation the methodology.

4. Limitations of the Analysis and Evaluation

- The evaluation presented in Figure 9 could be moved to the appendix as this analysis does not really focus on the high-resolution product and mainly serves as a sanity check, since the monthly GRACE Mascon dynamics are imposed by construction on the downscaled product. Therefore trends and dynamics at monthly and longer time scales are linked to the original GRACE data.

- A disentanglement of the FLDAS-CA bias into its individual components (E, P, and R) could strengthen the analysis by highlighting the sensitivity to each hydrological variable.

- While the evaluation against the ITSG-2018 dataset is highly valuable (particularly Figure 6, which shows that downscaling relying on satellite observations (MIX) can recover sub-weekly processes (correlation > 0.6)), I encourage adding an indirect evaluation using soil moisture data for three reasons:

1. Blank et al. (2023, https://doi.org/10.5194/hess-27-2413-2023) demonstrated that HR GRACE products provide insights into subsurface water storage dynamics. Exploring similar relationships with downscaled TWSC would be highly informative.

2. Recent soil moisture products are now available at daily, 1 km resolution (e.g., Brocca et al., 2025; https://doi.org/10.1016/j.scitotenv.2024.174087), enabling co-evaluation at comparable spatio-temporal scales.

3. Soil moisture evaluation could help disentangle precipitation-driven and irrigation-driven storage changes, particularly relevant for the Fergana Valley.

- The evaluation based on precipitation events should be presented in a dedicated subsection, as it is currently placed within the irrigated-area section without a clear link. To better address the main objective of the paper, I recommend comparing downscaling results obtained with different precipitation forcings and explicitly showing how precipitation errors propagate into HR TWSC estimates. Including precipitation time series in Figure 12 would greatly strengthen this discussion. Finally, if river discharge observations are available for this period and for these rivers, they would further strengthen the analysis by showing that in situ surface water observations capture this additional amount of water.

**Specific Comments**

- L41: The native GRACE spatial resolution is approximately 300 km.
- L57: Kalu et al., 2024 (not 2014).
- L139: "00:00 record" ?
- L190: In Eq. (1), reverting to TWS in the central difference is confusing. Since absolute TWS is not observable from GRACE, the last equality can be removed, and TWSA should remain the variable of interest.
- L210: Eq. (4) appears to compute TWS rather than TWSC. Similar ambiguities between S and dS occur elsewhere and should be systematically corrected.
- Figure 3:
    - The top-right box ("GRACE- vs WB-based") should be placed after the downscaling steps for consistency with the text.
    - The box should refer to WB-based S (prior to central differentiation).
    - Use NSE instead of correlation and RMSE, to be consistent with Figure 8.
    - Use consistent notation (S/dS or TWSA/TWSC) throughout the figure.
- Figure 4:
    - Y-axis label should be TWSC or dS, not dTWS/dt (same comment applies to other figures).
    - Avoid visual linear interpolation in periods with missing data (e.g., 2017).
- Figure 10b: The title should refer to TWSC, not TWS.